# Sustainable Resource Allocation and Reduce Latency Based on Federated-Learning-Enabled Digital Twin in IoT Devices

**DOI:** 10.3390/s23167262

**Published:** 2023-08-18

**Authors:** Mohammed A. Alhartomi, Adeeb Salh, Lukman Audah, Saeed Alzahrani, Ahmed Alzahmi, Mohammad R. Altimania, Abdulaziz Alotaibi, Ruwaybih Alsulami, Omar Al-Hartomy

**Affiliations:** 1Department of Electrical Engineering, University of Tabuk, Tabuk 71491, Saudi Arabia; 2Faculty of Information and Communication Technology, University Tunku Abdul Rahman (UTAR), Kampar 31900, Malaysia; 3Faculty of Electrical and Electronic Engineering, Universiti Tun Hussein Onn Malaysia, Batu Pahat 86400, Malaysia; 4Department of Industrial Engineering, University of Tabuk, Tabuk 71491, Saudi Arabia; 5Department of Electrical Engineering, Umm Al-Qura University Makkah, Mecca 24382, Saudi Arabia; 6Department of Physics, Faculty of Science, King Abdulaziz University, Jeddah 21589, Saudi Arabia

**Keywords:** internet of things, digital twins, resource allocation, energy consumption, deep-RL

## Abstract

In this article, we utilize Digital Twins (DT) with edge networks using blockchain technology for reliable real-time data processing and provide a secure, scalable solution to bridge the gap between physical edge networks and digital systems. Then, we suggest a Federated Learning (FL) framework for collaborative computing that runs on a blockchain and is powered by the DT edge network. This framework increases data privacy while enhancing system security and reliability. The provision of sustainable Resource Allocation (RA) and ensure real-time data-processing interaction between Internet of Things (IoT) devices and edge servers depends on a balance between system latency and Energy Consumption (EC) based on the proposed DT-empowered Deep Reinforcement Learning (Deep-RL) agent. The Deep-RL agent evaluates the performance action based on RA actions in DT to distribute its bandwidth resources to IoT devices based on iteration and the actions taken to generate the best policy and enhance learning efficiency at every step. The simulation results show that the proposed Deep-RL-agent-based DT is able to exploit the best policy, select 47.5% of computing activities that are to be carried out locally with 1 MHz bandwidth and minimize the weighted cost of the transmission policy of edge-computing strategies.

## 1. Introduction

Beyond Fifth Generation (B5G) architecture with Digital Twin (DT) is anticipated to evolve in the network autonomy and generative intelligence properties, which could optimize and adapt the network. The concept of a DT can be explored to effectively enable the properties of B5G wireless systems. Enhancing the quality of applications and the user experience of services such as autonomous vehicles and smart cities, in practice, depends on evaluating and mining data from the edge network by allocating limited resources and optimizing the network to deliver high-quality services [1,2]. The DT paradigm is one of the most exciting technologies, which can offer instantaneous wireless connectivity and very reliable wireless communication in a B5G network [3]. Due to the huge volume of sent data and the distance between end users and the server, the DT wireless networks are proposed to minimize latency and improve reliability for edge-computing applications [4]. Despite these recent advances and supporting technologies, it is still difficult to create robust control algorithms for physical systems because of the discrepancy between the results of data analysis and the required level of rapid data on their physical systems’ conditions [5]. At present, the centralized computing model imposes a significant burden on communication, which also raises concerns about data security. The Internet of Things (IoT) framework’s provision of intelligent services is enabled by data processing based on Artificial Intelligence (AI) algorithms. In IoT networks, the server can gather the operational states and create behavior models for various environmental conditions to create DTs. Solving this problem relies on machine learning techniques being applied to distributed end IoT devices to establish a DT network at the edge. In order to enable secure collaborative learning and develop trust among untrusting users, blockchain-enabled Federated Learning (FL) is capable of improving reliability and enhancing data security in a network. The proposed edge intelligence and a blockchain-powered IoT increase service capacity while lowering edge service costs by utilizing cross-domain sharing and a credit-differentiated transaction approval process [6]. Moreover, the distributed content caching system integrates blockchain technology to improve the efficiency of distributed learning by addressing the problem of data privacy with high-dimensional and time-varying characteristics based on the proposed Deep Reinforcement Learning (Deep-RL) [7].

### 1.1. Related Works

Building DT models requires the synchronization of a massive amount of data, but the digitization of the IoT is hampered by the limited computing power and communication ability. Our proposed strategy is a framework based on a number of innovative and beneficial technologies and algorithms, such as the incorporation of a blockchain and FL into the DT for edge networks. Many forms of data from a physical entity can be integrated into a digital space thanks to a DT [8]. The efficiency of DT was determined by the analysis of a sizable amount of data created and employed in a variety of applications, such as intelligent scheduling in smart cities, wireless network resources, and industrial real-time monitoring and optimization [9,10]. Some works concentrated on the subject of wireless networks. For instance, a DT edge network was proposed by the authors of [11] to provide a blockchain-enabled Resource Allocation (RA) and scheduling relaying users’ manner. To reduce the effects of unreliable communication between users and edge servers, the authors in [12] proposed a DT wireless network to reduce latency and improve reliability for edge-computing applications by mapping IoT devices to DTs in edge servers to improve the efficiency of AI algorithms. In addition, combining the strengths of blockchain and AI has led to significant advances in the delivery of secure and efficient resource management in wireless networks. By integrating AI and blockchain, the authors of [13] suggested a reliable and intelligent resource-sharing system to further build a content caching system using Deep-RL. The authors in [14] proposed a support Vector Machine Training (SVM) technique that protects privacy, called safe SVM, and used blockchain technology to create an IoT network data-sharing framework that is reliable and secure for use by multiple data providers. It is difficult to figure out how to interpret such data using IoT devices with limited resources. The digitalization of the IoT is restricted by the limited computing power and communication abilities, which makes it difficult to adopt blockchain to develop DT models that demand trust and consensus across distributed users. Additionally, blockchain technology enhances distributed learning’s effectiveness in solving the problem of data privacy with high-dimensional and time-varying attributes based on the provided Deep-RL.

FL is a new paradigm that is attracting a lot of attention. It offers a brand-new distributed machine learning approach that can reduce the risk of information leaks and hence increase data privacy [15]. As the wireless networks research community’s interest in FL grows, a significant amount of effort [16] has been made in this field to enhance FL performance through resource optimization. Based on an empowered FL scheme for DT-edge networks to improve user learning security and data privacy, the authors in [17] presented an FL and permissioned blockchain for DT-edge networks to improve effective communication and information security for IoT applications. To guarantee learning accuracy, reliability, and security, the other authors proposed a dual-driven learning solution for the DT-IoT, enabled by blockchain, to ensure real-time interactions for sustainable computing. This solution depends on combining the DT with an edge network and adopting blockchain technology [18]. However, the authors of [19] proposed using FL in wireless networks and offered the best compromise between Energy Consumption (EC) and the cost of learning time. In other words, the security of FL has also been discussed in terms of users’ data privacy and gradient leakage [20,21]. To increase privacy regarding the loss boundaries in FL and improve security and privacy, the authors in [22] proposed implementing Bayesian differential privacy by an effective method that enables a learning agent to modify its policy and improve data-processing efficiency. Th method proposed in [23] applies a new paradigm of DT networks by applying Deep RL. Most recent works [14,15,16] employ blockchain to build consensus and trust among dispersed users, and guarantee the security and privacy of user data by establishing a distributed ledger amongst users. To fill this gap, this work focuses on achieving real-time data processing and computing to the edge plane, based on the introduction of DT wireless networks, by integrating DTs into wireless networks. Other studies [18,19,20,21,22,23] did not focus on RA’s relationship with user scheduling and bandwidth allocation in IoT devices based on the use of Deep-RL in DTs to evaluate its performance. Moreover, the above studies [18,19,20,21,22,23] have not investigated sustainable RA or the optimized data relay challenge, wherein Deep Neural Networks (DNN) are harnessed as the strategic schedulers within the strategy scheduler in the recommended solution to balance learning accuracy and time expenditure. The significant contribution of this framework is to establish an enduring RA system, guaranteeing seamless real-time communication between IoT devices and edge servers. This accomplishment hinges on the formulation of an optimized data relay challenge, wherein DNNs are harnessed as the strategic schedulers within the suggested approach. The objective is to strike a harmonious equilibrium between system responsiveness and EC, orchestrated by the innovative utilization of the Deep-RL agent empowered by the proposed DT. The Deep-RL agent evaluates the efficiency of an action by considering RA actions within the DT. According to the iterations and actions taken, this assessment allocates bandwidth resources to IoT devices. The main goal is to come up with an ideal policy that increases learning effectiveness at each stage of the procedure. Then, for collaborative computing, we proposed a blockchain-enabled FL framework running in the DT wireless networks to increase data privacy while enhancing system security and reliability.

### 1.2. Motivation and Contributions

To increase the IoT devices’ performance, and boost the system’s reliability, security, and enhance data privacy, we formulated the DT edge networks’ model optimization problem and propose a blockchain-enabled FL framework to improve communication efficiency. Accordingly, by simultaneously considering DT association, training data batch size, and bandwidth allotment, we design the data-relaying optimization problem and employ Deep Neural Networks (DNN) as the strategy scheduler in the recommended solution to balance learning accuracy and time expenditure. The main contributions of this paper are as follows: To effectively and appropriately optimize IoT networks, we proposed a DT-empowered IoT framework that maps a data-driven DT device’s real-time operation using blockchain technology.We use the FL framework to build the DT edge network models by employing a gradient descent approach that can lower the overhead of data transfer and safeguard data privacy. Furthermore, we use asynchronous model aggregation to increase communication efficiency, which depends on enhancing the target of local computing by decreasing wait times and keeping track of the training process achievement at edge servers to reduce redundant user delays.We present a unique blockchain-supported DT-IoT framework to reduce the system delay and EC and provide secure and reliable computing in DNN, as well as new insight into the impact of the training process achievement requirements on the RA efficiency. The proposed Deep-RL agent based on DT evaluates the performance action based on RA for the user scheduling and bandwidth allocation in IoT devices in order to increase system stability, develop a balance in learning accuracy, and guarantee the learning accuracy of IoT devices.

## 2. Materials and Methods

In this study, we present a DT-IoT system that can be integrated with blockchains and users for edge computing. We suggest a blockchain-powered FL architecture to increase security and guarantee the performance of edge computing. It has a user plane and an edge plane that incorporate the DT into the edge network. To achieve secure-aware and reliable-aware edge intelligence, the proposed approach integrates the blockchain and DT based on an increase in the output accuracy and reduction in the loss in terms of DNN in IoT systems. IoT devices receive the aggregated data from DT models following the blockchain consensus process. In order to achieve sustainable RA in the IoT and address the issue of poor accuracy, operational IoT devices can submit data that are sent to edge servers for real-time updates. We set IoT device using the notation ҡ ={1, 2, …,Ҡ}. We consider a set ҡ of Ҡ IoT devices in the user layer and connect them to the edge plane via wireless communications. The edge layer is made up of multiple edge servers that have base stations equipped with mobile edge servers. Wireless communications are used by each edge server with the j={1, 2, …,J } to communicate with the UEs within its coverage. To ensure secure and reliable data transmission, edge servers are managed as blockchain nodes. To protect real-time IoT devices, edge servers build DTs on the edge plane [18]. The IoT device’s data are pre-processed to obtain vectors that reflect the operating state before being used in the real-time twin modeling process. Then, edge servers collect and process IoT device operating states to produce DT models, which are indicated as follows:(1)$$Ɛ=\left({ƿ}_{ҡ},{Ʀ}_{ҡ},{Ɗ}_{ҡ},{Ƒ}_{ҡ},\varphi_{ҡ}\right),\;$$
where ${ƿ}_{ҡ}$ represents the IoT devices’ transmission power, ${Ʀ}_{ҡ}$ represents their upload data rate, ${Ɗ}_{ҡ}$ represents their pre-processed data set, ${Ƒ}_{ҡ}$ represents their processing capacity, and $\varphi_{ҡ}$ represents the performance index weight parameter of the improvement target in the DT-IoT system.

### 2.1. Sustainable Blockchain Model for Secure Communication

Data exchange is becoming an essential element of the IoT for DTs and is essential to maintaining IoT security. Reducing the duration of time required for model training in various applications is a crucial challenge in the B5G network due to the expansion of the user devices, the demand for communication with ultra-low latency, and the dynamic network condition. IoT devices communicate data to edge servers in the DT-IoT system enabled by blockchain via orthogonal frequency division multiple access (OFDMA). To transmit data, sub-channels Ƈ are shared with the IoT device ҡ. The maximal data rate of an IoT device ҡ is written as
(2)$${Ʀ}_{ҡ}\left(\tau \right)=\sum_{Ƈ=1}^{Ƈ}\beta \zeta_{ҡ,Ƈ}\log_{2}\left(\frac{{𝒽}_{ҡ,Ƈ}\left(\tau \right)\;{𝒫}_{ҡ,Ƈ}\left(\tau \right)}{{Ŋ}_{0}}\right),$$
where β represents the transmission bandwidth, ${𝒽}_{ҡ,Ƈ}\left(\tau \right)$ represents the IoT device’s ҡ for channel gain at time slot τ, $\zeta_{ҡ,Ƈ}$ represents the number of sub-channels allocated to IoT device ҡ, ${𝒫}_{ҡ,Ƈ}\left(\tau \right)$ is the IoT device’s ҡ for transmission power in subchannel Ƈ, and ${Ŋ}_{0}$ is the noise power. The total time delay that an IoT device’s DT ҡ takes to update its status is denoted as
(3)$${Ʈ}_{upd}=\frac{\left|{Ɗ}_{ҡ}\left(\tau \right)\right|}{\;{Ʀ}_{ҡ}\left(\tau \right)}.\;$$

The time of data upload depends on a blockchain delay and a communication delay in order to reduce the transmission load as follows: 

**Firstly**: The blockchain can significantly boost the cost-effectiveness of DTs by updating real-time data. Based on an analysis of the consensus procedure of blockchain nodes, the blockchain latency between edge servers consists of the time it takes for information to spread across edge servers and for new blocks to be created ${Ʈ}_{bd}$.
(4)$${Ʈ}_{bd}=\underset{j}{\max}\;\log_{2}\;N\left\{\frac{{Ɗ}_{ҡ}{Ʈ}_{j}\left|{ϣ}_{e}\right|}{{Ʀ}_{es}}\right\},$$
where N represents the number of edge servers, $\left|{ϣ}_{e}\right|$ is the level of transmitted model parameters of ҡ, and ${Ʀ}_{es}$ represents the achievable data transmission between edge servers. 

**Secondly**: The time for uploaded data depends on communication delays, which depends on the data size of DT ${Ƣ}_{i}$ stored by edge servers, ${⨍}_{j,\;{Ƣ}_{i}}^{C}$, which represent the computation speed of the CPU cycle frequency at edge servers j, and $C_{j}$ represents the number of CPU cycles consumed to process an edge server to train a data sample. The CPU-cycle frequencies enable operation at various frequencies to regulate power consumption, which helps to shorten the IoT device communication’s computation times and speed up the FL training process. The communication delay can be written as
(5)$${Ʈ}_{cd}=\underset{j}{\max}\sum_{i=1}^{{Ƣ}_{j}}\frac{{Ɗ}_{{Ƣ}_{j}}\;}{{⨍}_{j,\;{Ƣ}_{i}}^{C}}C_{j}.$$

From the analysis above, the total delay is denoted as
(6)$${Ʈ}_{tot}={Ʈ}_{bd}+{Ʈ}_{cd},$$
where |${Ɗ}_{ҡ}\left(\tau \right)$| represents the data size for IoT device updates.

### 2.2. Learning Accuracy for DT-IoT -FL

The edge association problem is crucial for minimizing the total time cost in DT edge networks while maintaining the required level of learning accuracy. To evaluate the decision-making capabilities of DTs in our proposed IoT networks, FL is utilized for traffic load prediction and training independent local models based on the locally collected data and then shares their model parameters on the edge servers using wireless transmitting data [17,24]. Blockchain-enabled FL is capable of increasing reliability and enhancing data security in a network to enable secure collaborative learning and foster trust among untrusted users. The proposed method incorporates blockchain and DT to improve output accuracy and lower loss, to create secure-aware and reliable-aware edge intelligence. Due to the proliferation of user devices, it is necessary to shorten the amount of time needed for model training in the various applications, as illustrated in (2) and (3), in order to maintain secure connections. Depending on the particular components of latency being assessed, the dataset utilized for latency calculations in a blockchain setting can change. Reductions in the amount of time needed for model training across a variety of applications due to the growth in user devices and the necessity of ultra-low latency connection are shown in (3). The amount of time it takes to upload data is influenced by the blockchain latency (4) between edge servers and the communication delay for the data size of DT ${Ƣ}_{i}$ that reflects the CPU cycle frequency ${⨍}_{j,\;{Ƣ}_{i}}^{C}$ (5). The local dataset of device ҡ is ${Ɗ}_{ҡ}$, which is a collection of data samples with the coordinates $\{{Ƨ}_{j}$, ${Ȥ}_{j}{\}}_{j=1}$, where ${Ƨ}_{j}$ is sample j−th input and ${Ȥ}_{j}\;$is sample j−th output. Using the data set from IoT device ҡ, the local loss function can be calculated as
(7)$${Ƒ}_{ҡ}\left(ϣ\right)=\frac{1}{{Ɗ}_{ҡ}}\sum_{j=1}^{J}\sum_{ҡ\in {Ɗ}_{ҡ}}{\;⨍}_{ҡ}\left({ϣ}_{j},\;{Ƨ}_{jҡ},\;{Ȥ}_{jҡ}\right),\;\forall \;ҡ\in Ҡ$$
where ${Ɗ}_{ҡ}=\left|{Ɗ}_{ҡ}\right|$ represents the number of collected data samples by the IoT device, and ${\;⨍}_{ҡ}\left(ϣ\right)$ is the loss function that calculates the error in the local training model for data sample j. In order to train model parameters, we employ a gradient descent approach. When iteration begins, edge servers provide the same knowledge to all IoT devices τ. To train the model parameter ϣ(τ), each IoT device integrates its local dataset ${Ɗ}_{ҡ}$, denoted as:(8)$${ϣ}_{j}\left(\tau \right)=ϣ\left(\tau -1\right)-ƛ\nabla {Ƒ}_{ҡ}ϣ\left(\tau -1\right),$$
where ƛ>0 is the learning step size, and $\nabla {Ƒ}_{ҡ}ϣ\left(\tau -1\right)$ indicates the gradient value of the parameter’s loss function ϣ(τ−1). To improve learning accuracy, reducing the EC and URLLC of the DT-IoT system depends on enhancing the target of local computing by decreasing wait times and minimizing the communication load. This depends on keeping track of the training process achievement at edge servers to maintain a global iteration that records the aggregation process’ performance in the blockchain. The IoT devices compute updated ${ϣ}_{j}\left(\tau \right)$ in the subsequent iterations by computing the gradient descent of the local loss function in iteration τ. The size of the local training data and the loss value, represented by $\ell =\underset{ҡ\in Ҡ}{\min}\;{Ɗ}_{ҡ}/{Ƒ}_{ҡ}\left(ϣ\right)$, are used to determine the learning accuracy for each IoT device.

### 2.3. IoT Device for EC

In this section, local training and data transfer are the two key steps of the EC. The computational EC of CPU frequency generated by IoT devices ҡ is denoted by ${⨍}_{ҡ}^{C}$ [20,25]. For local computation, the EC of device ҡ can be expressed as follows:(9)$${Ԑ}_{ҡ}^{c}=\delta \;C_{ҡ}\;\left|{Ɗ}_{ҡ}\right|{\left({⨍}_{ҡ}\right)}^{2},$$
where $C_{ҡ}$ is the total amount of processing CPU cycles for IoT devices to train a data sample, $\delta =\gamma \log_{2}(\frac{1}{ѱ})$ represents the effective switched capacitance, γ represents a constant related to the data size and ѱ  is the minimum loss at this rate. The local model upload requires the following EC:(10)$${Ԑ}_{ҡ}=\frac{{ƿ}_{ҡ}\;\left(\tau \right)\;{Ɗ}_{ҡ}\left(\tau \right)}{\beta \;\zeta_{ҡ,Ƈ}\log_{2}\left(\frac{{𝒽}_{ҡ,Ƈ}\left(\tau \right)\;{𝒫}_{ҡ,Ƈ}\left(\tau \right)}{{Ŋ}_{0}}\right)}.$$

The amount of energy consumed in terms of DT when using an edge server can be written as:(11)$${Ԑ}_{j}^{c}\;=\delta \;C_{j}\;\sum_{i=1}^{{Ƣ}_{j}}{Ɗ}_{{Ƣ}_{j}}\;{\left({⨍}_{j,\;{Ƣ}_{i}}^{C}\right)}^{2}.$$

The total EC of the DT-IoT devices connected to FL can be calculated as:(12)$${Ԑ}_{total}=\sum_{ҡ=1}^{Ҡ}\left({Ԑ}_{ҡ}^{c}+{Ԑ}_{ҡ}\right)+\sum_{j=1}^{J}\left({Ԑ}_{j}^{c}\right).$$

## 3. Formulation of the Communication Effectiveness Problem for DT-IoT Using FL

In this section, our goal is to develop a federated model that minimizes the weighted cost and enhances learning accuracy from distributed IoT devices. So, the FL model must be trained with minimal resource consumption due to the limited computing and communication resources of IoT devices. A trade-off between learning accuracy and resource efficiency is established by the combinatorial problem of resource optimization. The optimization problem can be expressed as: (13)$$\underset{⨍,\;{Ƣ}_{i},𝒪}{\min\;\frac{1}{\left|Ɗ\right|}}\sum_{j=1}^{J}\sum_{ҡ=1}^{Ҡ}{⨍}_{ҡ}\left({ϣ}_{j},\;{Ƨ}_{jҡ},\;{Ȥ}_{jҡ}\right)+\sigma \left({Ʈ}_{tot}+{Ԑ}_{total}\right)$$
(13a)s. t. 𝒪∈{0, 1}, 
(13b)$$\sum_{i=1}^{{Ƣ}_{j}}\;{Ƣ}_{i}=Ҡ,\;$$
(13c)$${⨍}_{ҡ}^{min}\;{⨍}_{j,\;{Ƣ}_{i}}^{C}\leq {⨍}_{ҡ}\leq \;{⨍}_{j}^{max}\;{⨍}_{ҡ}^{max},\;\forall \;ҡ\;\in \;Ҡ$$
(13d)$${𝒫}_{ҡ}^{min}\leq {𝒫}_{ҡ}\leq {𝒫}_{ҡ}^{max},\;\forall \;ҡ\;\in \;Ҡ$$
(13e)0≤σ≤1, 
where 𝒪  is the controlling factor that determines how the loss function and cost functions are balanced, as shown in (13a). The time for uploaded data depends on a number of DT ${Ƣ}_{i}$ that represent the IoT device behaviour model created by examining their previous running data during the time slots given to IoT devices at iteration τ by DT ${Ƣ}_{i}\left(\tau \right)$. The constraint (13c) represents the computing resources allocated for cycle CPU frequency, which are less than the upper or maximum bounds ${⨍}_{j}^{C}$ and can be solved according to [23]. From (13d), edge servers are shown to have a sufficient power supply compared to IoT devices. From (13e), the scheduling policy vector for IoT devices σ is shown to be able to determine which relay IoT devices will send the nearest training parameter. The transmission time for uploaded data is determined by a number of ${Ƣ}_{i}$ that indicate the behaviour model developed by analysing their historical operating data throughout the time slots provided at iteration by ${Ƣ}_{i}\left(\tau \right)$. The restriction (13c) denotes the computational resources assigned to cycle CPU frequency, which are fewer than the upper or maximum constraints ${⨍}_{j}^{C}$. In contrast to IoT devices, edge servers from (13d) have a suitable power. As seen in (13c) and (13d), when resources are limited, blockchain technology can be included in both edge and IoT devices. It is difficult to use conventional convex optimization algorithms to successfully solve problems in (13), which is an NP-hard optimization issue. Updates to the global model depend on the minimization of the problem in (13) $\frac{1}{{Ɗ}_{ҡ}}\sum_{j=1}^{J}\sum_{ҡ\in {Ɗ}_{Ҡ}}\;{⨍}_{ҡ}\left({ϣ}_{j},\;{Ƨ}_{jҡ},\;{Ȥ}_{jҡ}\right)$ according to the gradient descent algorithm used to address ${Ƒ}_{ҡ}\left(ϣ\right)$, and the ability to obtain the exact learning used on training data and the accuracy of the local data. The objective of communication improvements to edge servers is explained by the equation in the second term (13), as follows: (14)$$\underset{{Ƣ}_{i},\;𝒪\;}{\min}\left({Ʈ}_{tot}+{Ԑ}_{total}\right)$$
(14a)s. t. 𝒪∈{0, 1}, 
(14b)$$\sum_{i=1}^{{Ƣ}_{j}}\;{Ƣ}_{i}=Ҡ,\;$$
(14c)$${Ʈ}_{tot}\leq {Ʈ}_{th},$$
(14d)$${Ԑ}_{total}\;\leq \;{Ԑ}_{th},$$
where $\;{Ʈ}_{th}$ and ${Ԑ}_{th}$ represent the estimated threshold for all IoT devices in the DT for delay and EC respectively. As all edge servers have DT models $\;{Ƣ}_{i}$, the number of IoT devices is equal to the number of edge servers according to (14b). Maintaining DTs and training data in real-time would be difficult because of the predicted growth in IoT devices and data traffic in B5G networks, which means that providing sustainable RA and processing large amounts of data become considerably more complicated. Each device needs to reduce the EC of devices and the costs of system delays. In addition, the efficiency of the learning accuracy and sustainability of the DT-IoT system should be guaranteed by maintaining an appropriate level of URLLC and ensuring the learning accuracy of IoT devices. In order to provide sustainable intelligent contact between IoT devices and edge servers, and to increase the effectiveness of integrated DTs with edge networks, real-time data processing must strike a balance between system delays and EC. To provide sustainable RA, we proposed a DT-empowered Deep-RL-based algorithm to train DNN and generate the policy for each IoT device.

### 3.1. DNN-Train-Based Resource Scheduling Algorithm in DT-IoT

The DNN can explore the DT to avoid the loss of training caused by exploration-intensive resource consumption in real-time based on transmitting the “straggler’s” transmission tasks to the users with higher connectivity. The model ${ϣ}_{j}$ that was trained by ҡ has insufficient communication abilities; to improve the communication efficiency, we propose an IoT device ӄ that might be used to relay the transmission activities of the model ҡ. The potential IoT devices ӄ can perform all the transmission requirements of ҡ to determine whether ${ϣ}_{j}$ should be transmitted from ҡ to ӄ, and decide on the responsibility of the scheduling policy to obtain the value of 𝒪∈{0, 1}. After that, IoT devices with the ${Ƣ}_{i}$ designation are provided with bandwidth resources based on their statuses. Using DNN, IoT devices can become more responsive to their surroundings and use less network capacity by uploading fewer pieces of data to edge servers and saving valuable network bandwidth [26,27]. The DNN is represented as $\phi =\left\{{Ɯ}^{\ell },{ɓ}^{\ell }\right\}$, where ℓ∈{1,….,ℒ} represents the DNN’s ℓ−th layer. Using the weight vectors ${Ɯ}^{\ell }$ and bias in vector ${ɓ}^{\ell }$, the DNN can be written as:(15)$${Ȥ}^{\ell }=⨍\left({Ɯ}^{\ell }{Ƴ}^{\ell }+{ɓ}^{\ell }\right).$$

Let ${Ƴ}^{\ell }$ represent the input to the  ℓ−th layer of the network, and ${Ȥ}^{\ell }$ is the output of neuron ᴎ in layer l. The system’s states, such as the transmission rate that can be achieved and the computational power of each IoT device, are input into the DNN during the learning process. The output of the DNN is the best relay policy. The DNN models are trained using the training data in each epoch (Ʀ, ⨍, Ƣ, Ʈ). In order to reduce the training complexity in DNN, we proposed using a Deep-RL agent to train the policy to reduce the scale of training data, so that they are be smaller in volume and resource-efficient.

### 3.2. Deep-RL Agent for RA

Reinforcement learning has been successfully used to handle RA and task resource scheduling problems in the DT-IoT. Deep-RL evaluates the performance action based on RA actions (${𝒪}_{\tau }$, ${Ƣ}_{\tau }$), where each agent is optimally assigned in DT to distribute its bandwidth resources to IoT devices based on iteration and the actions taken to distribute resources (${𝒪}_{\tau }$, ${Ƣ}_{\tau }$). Thus, the action-value function Ѧ, state-space δ, and reward K of the Deep-RL framework are all explicitly defined. Furthermore, the Deep-RL framework takes the required action $a_{\tau }\in \;Ѧ$, which consists of bandwidth RA Ƣ and IoT devices scheduling 𝒪 at every state $s_{\tau }\in \delta$. The state environment can be defined as $s_{\tau }=\left\{{Ƒ}_{\tau },\;{⨍}_{\tau },{Ʀ}_{\tau },{ϣ}_{\tau }\right\},$ where ${Ƒ}_{\tau }$ represents loss value, ${⨍}_{\tau }$ is the speed of the CPU cycle frequency at edge servers, ${Ʀ}_{\tau }$ is the data rate vector that can be achieved and regulated by the bandwidth allocation policy, and$\;{ϣ}_{\tau }$ represents the learning for IoT device [25,28]. To achieve high efficiency in the learning accuracy and sustainability of the DT-IoT, the agent continues to the next state and receives a reward immediately. The reward function $K_{\tau }$ can be defined as:(16)$$K_{\tau }=\{\begin{array}{c} ƛ{Ʈ}_{tot}+C\;\;\;\;\;\;\;\;\;\;if\;\;\;⨍,\;\;P,\;Ԑ\;\leq \;{Ԑ}_{total}\;\\ ƛ{Ʈ}_{tot}-C\;\;\;\;\;\;\;\;\;\;\;\;\;\;\;\;\;\;\;\;\;\;\;\;\;\;\mathrm{otherwise} \end{array}\;,$$
where C represents the number of CPU cycles executed to train data for IoT devices. From (16), a positive reward $ƛ{Ʈ}_{tot}+C$ will be added and encouraged as a suitable trade-off to balance the RA between computing and communication resources if all metrics following the action pass the constraint check. Using a positive reward $ƛ{Ʈ}_{tot}+C$ in combination with an active learning technique can greatly reduce the cost of training a model. Based on the current states obtained from DT, the performance of the action is quantified by the reward function at the end of the iteration. The constraints in (16) $if\;Ԑ\;\leq \;{Ԑ}_{total}$ will also reduce system time costs and every agent attempts to utilize the optimum policy to maximize an accumulative reward based on frequently updated 𝒪 and Ƣ, minimizing the weighted cost of the transmission policy and enhancing learning efficiency at every step. Otherwise, when the training sample for CPU C was negative, the iteration completion time and the reward K(τ) have a negative relationship. To evaluate and improve the reward function of in DT-IoT, the future cumulative discounted reward at a time slot can be defined as follows:(17)$${𝔯}_{ҡ,\tau }=K_{\tau +1}+\emptyset K_{\tau +2}+\cdots =\sum_{\tau =1}^{Ʈ-1}{Ӷ}^{ҡ}K_{\tau +ҡ+1}\;,$$
where ∅∈[0, 1] represents the discount factor. Based on the appropriate running states from the DT-enabled exploration, the DNN experience training data are obtained $s_{\tau }=\left\{{Ƒ}_{\tau },\;{⨍}_{\tau },{Ʀ}_{\tau },{ϣ}_{\tau }\right\}$. The final output is obtained by applying the activation function ⨍ to ${Ƴ}^{\ell }$ in the hidden layers using the Rectified Linear Unit $ReLU\;\left({Ƴ}^{\ell }\right)=max\left(0,\;{Ƴ}^{\ell }\right)$. Applying the activation function ⨍  to Ƴ^ℓ^ in the hidden layers using the Rectified Linear Unit $ReLU\;\left({Ƴ}^{\ell }\right)=max\left(0,\;{Ƴ}^{\ell }\right)\;$yielded the desired output [29]. Using the states $s_{\tau }$ as the input to DNN, the vectors of 𝒪 and Ƣ are the output for the relaying strategies towards network states, which are investigated and generated as: (18)$$𝒪=\sum_{\ell =1}^{ℒ}⨍\left({Ɯ}^{\ell }{Ƴ}^{\ell }+{ɓ}^{\ell }\right).$$

Minimizing the weighted cost of transmission policy for distributed IoT devices in (13) depends on adjusting the values of ${𝒪}_{jҡ}\in 𝒪,$ assigning the relevant transmission time ${Ƣ}_{j}$ and saving this into replay memory as training samples when $\;{Ʀ}_{ҡ}\leq {Ʀ}_{j}$ and ${⨍}_{ҡ}\;\geq {⨍}_{j}$. The minimization of the cost necessary to provide the optimal relaying strategy for state $s_{\tau }$ can be achieved as follows:(19)$$𝒪\underset{{𝒪}_{ҡj}}{\max}\sum_{ҡ}^{Ҡ}\sum_{j}^{J}{𝒪}_{jҡ}\times {Ƣ}_{j}\times \frac{C}{{⨍}_{jҡ}\times {Ʀ}_{jҡ}}\;s.t.\;{𝒪}_{jҡ}\in \left\{0,\;1\right\},\;\sum_{i=1}^{{Ƣ}_{j}}\;{Ƣ}_{i}=Ҡ,\;{𝒫}_{ҡ}\leq {𝒪}_{jҡ}\times {Ƣ}_{jҡ}\times {⨍}_{jҡ},\;{𝒫}_{ҡ}^{min}\leq {𝒪}_{jҡ}\times {Ƣ}_{jҡ}\times {⨍}_{jҡ}\leq {𝒫}_{ҡ}^{max},$$

In (19), every agent attempts to utilize the optimum policy to maximize the accumulative reward based on frequently updated 𝒪 and Ƣ, minimizing the weighted cost of transmission policy and enhancing learning efficiency at every step, as shown in Algorithm 1 and Figure 1 [12,20].
**Algorithm 1: Deep-RL Agent for RA Scheduling Algorithm in DT-IoT**Input Set Ɯ,ɓ, 𝒪$,\;{𝒫}_{ҡ}$$,\;{⨍}_{ҡ}$and Ƣ;Output: value 𝒪 satisfying (19);For each edge server j≤J, complete the following;Initialize DT-IoT environment setup;For each time slot τ, complete the following;$\mathrm{Evaluate}\;\mathrm{the}\;\mathrm{performance}\;\mathrm{action}\;\mathrm{based}\;\mathrm{on}\;\mathrm{RA}\;\mathrm{actions}\;({𝒪}_{\tau }$$,\;{Ƣ}_{\tau }$);
Check the suitable trade-off to balance the RA between computing and communication according to (16);End for;Compute cumulative discounted reward at a time slot and transit to the next state $s_{\tau }$$\;\mathrm{in}\;\mathrm{DNN}\;\mathrm{based}\;\mathrm{on}\;\mathrm{the}\;\mathrm{actions}\;\mathrm{taken}\;\mathrm{to}\;\mathrm{distribute}\;\mathrm{resources}\;({𝒪}_{\tau }$$,\;{Ƣ}_{\tau }$);$\mathrm{Store}\;\mathrm{states}\;\mathrm{and}\;\mathrm{optimal}\;\mathrm{allocation}\;({𝒪}_{\tau }$$,\;{Ƣ}_{\tau }$$)\;\mathrm{into}\;\mathrm{replay}\;\mathrm{memory}\;\mathrm{when}\;{Ʀ}_{ҡ}\leq {Ʀ}_{j}$ $\mathrm{and}\;{⨍}_{ҡ}\;\geq {⨍}_{j}$;Minimize the cost necessary to obtain the optimal relaying strategy according to (18) and (19);End for;$\mathrm{Train}\;\mathrm{the}\;\mathrm{DNN}\;\mathrm{model}\;\mathrm{using}\;\mathrm{the}\;\mathrm{previously}\;\mathrm{saved}\;\mathrm{samples}\;(s_{\tau }$$,\;{𝒪}_{\tau }$).

## 4. Discussion

This section in the DT-IoT system proposes the evaluated methods for the Proposed Deep-RL agent based on the DT- RA algorithm. Here, a DT-IoT system with 4 edge servers and 16 IoT devices is built with help from the blockchain. We evenly divide the channel gain (${𝒽}_{ҡ,Ƈ}\left(\tau \right)$) into 11 levels and quantize it as [30] before determining the boundary values. FL efficiency is measured using the MNIST real-world dataset [31] and the Fashion-MNIST dataset [32]. Each dataset has a training set with 60,000 examples and a testing set with 10,000 examples as show in Table 1 [20,26].

In comparison to other methods, the proposed Deep-RL agent based on DT offers the best reward performance, quickest convergence, and most stable learning process. Exploring various data-relaying policies tells the agent how to maximize the overall reward, as shown in (16).

The size of the steps taken to update the model parameters during training is determined by the learning step size, which is an important variable shown in Figure 2. Convergence speed and stability are impacted. A new transaction is added to the blockchain every time a learning step involves changing the learning step size. The iteration number, learning step size, and possibly other pertinent metadata are all recorded in this transaction, along with other pertinent information. The best reward value for user scheduling and bandwidth allocation is attained when the learning accuracy is ƛ =0.001, more than when it is ƛ = 0.003. From Figure 2, the best reward is obtained when the learning strategy maximizes the learning accuracy through more than 4000 iterations. The stringent latency requirements of the increasing number of IoT devices may make delayed convergence impossible to achieve. As training iterations are extended, the worst performance only depends on the immediate reward based on RA for the user scheduling and bandwidth allocation, which increases overall loss.

Figure 3 shows the time cost related to the increasing number of iterations. The discount factor in Deep-RL, as seen in Figure 3, symbolises the preference for future benefits over present ones. Through the blockchain, transactions on a distributed network are verified and agreed upon. Due to the increase in training from ∅= 0.6 to ∅= 0.8, this process naturally generates time delays. The blockchain has the ability to contribute value in terms of data openness to avoid limiting the effectiveness of Deep-RL training. It is directly able to reduce the time required to train a Deep-RL agent. When the discount factor is ∅= 0.6, the time required to run our Deep-RL-based agent is substantially lower; this is because of the process of iterative exploration. Using a positive reward $ƛ{Ʈ}_{tot}+C$ in combination with an active learning technique can greatly reduce the cost of training a model. This will also reduce system time costs and promote appropriate trade-offs between RA and communication resources. To learn the best policies, the agents must interact with the environment for a long time. The overall time investment for training or testing in each iteration is comparable, despite occasional variations in the outcome curves, as shown in Figure 3. An increased discount factor somewhat results in an increase in time costs in each iteration; this is due to the fact that the policy-training procedure requires more computation when the discount factor is large.

In Figure 4, the proposed Deep-RL agent based on DT and the learning-agent-based random policy with random edge association are compared in terms of latency performance with respect to training rounds. Comparing the proposed Deep-RL agent to the RA strategy, it is found that the proposed Deep-RL agent greatly lowers the system time cost compared to the RA policy [D. Yueyue 2020]. In order to increase operational efficiency and decrease system latency for our scheme, the Deep-RL agent optimizes DT associations and optimally distributes communication resources. In addition, the learning agent based on random policy leads to a larger time cost compared to the proposed RA policy [D. Yueyue 2020]. The Deep-RL-agent-based DT can produce trained learning accuracy with good performance and use the best policy to maximize an accumulative reward based on an 𝒪 that is updated on a regular basis, and Ƣ that can make predictions on new or unseen data.

Figure 5 shows the highest achievable accuracy of the performance action based on RA actions in DT. Centralized training is the performance upper bound used to set the central trainer’s processing power to the total number of devices while ignoring the communication delay associated with collecting training data. The use of DT with Deep-RL agents offers good accuracy. This is because DT ideally assigns each agent a greater level of training to distribute bandwidth resources to IoT devices based on iteration and the actions carried out to distribute optimal resources allocation (${𝒪}_{\tau }$, ${Ƣ}_{\tau }$), which had a greater statistical power to more accurately and efficiently allocate resources. According to Figure 5, the proposed Deep-RL agent-based DT trade-off between the number of rounds and the latency per round is able to enhance the scheduling of more IoT devices and lower the number of rounds needed to achieve a fixed accuracy, but at the expense of a higher latency per round. From Figure 5, the latency per round can be decreased by scheduling fewer devices, but the convergence rate with respect to the number of rounds will be slower.

Figure 6 shows the total cost of the impact of transmission bandwidth. As bandwidth increases, the total cost declines, and the transmission times are shortened when the communication resource is sufficient. As the overall system bandwidth is increased, it is evident from the simulation results that all curves steadily decline. This is because the increasing bandwidth allocation has a substantial impact on increasing the data transmission rate between edge servers in DT and IoT devices. With the increased edge server processing capacity, the overall cost of edge computing strategies is greatly reduced. The proposed technique performs well when bandwidth is limited. When compared to the RA policy [D. Yueyue 2020] and the learning-agent-based random policy, respectively, our proposed algorithm achieves time reductions of 1.10 and 1.12, more than those obtained when the system bandwidth is 10 MHz. The proposed Deep-RL-agent-based DT selects 47.5% of computation processes to be carried out locally with 1 MHz bandwidth, compared to the RA strategy [D. Yueyue 2020], which only selects 43%.

From Figure 7, the efficiency of the proposed Deep-RL agent based on DT with RA policy [D. Yueyue 2020], the IoT edge resource is allocated based on the average processing demand of two services. When the edge resource is limited, the suggested Deep-RL agent based on DT outperforms the proposed RA policy [D. Yueyue 2020] by a wide margin. The performance advantage in lowering the service latency specifically drops from 1.26 at 1 GHz CPU frequency to only 0.56 at 1.2 GHz CPU frequency. The effect of the number of CPU cycles to be treated depends on the length of local model training, which was dependent on the number of CPU cycles; when many IoT devices are needed to fix the CPU frequency, the energy consumption significantly increases. The explanation is that an efficient RA is more crucial in settings where resources are scarce than it is in scenarios where resources are abundant. The outcomes verify the efficiency of the RA optimization subroutine for edge servers.

A system performance comparison is carried out between the proposed Deep-RL agent based on the DT algorithm and other algorithms to assess the effects of varying IoT device numbers. To confirm the effectiveness of trials, it is important to compare performance under various IoT device counts. Our suggested approach offers a larger reward, as seen in Figure 8. To achieve high-efficiency learning accuracy from (13c), (13d), the total energy cost ${Ԑ}_{total}$ is also checked as a budget that should not exceed an expected value $if\;⨍,\;𝒫,\;Ԑ\;\leq \;{Ԑ}_{total};$ then, the agent continues on to the next state and receives a reward immediately. In addition, the reward of the Deep-RL agent based on the DT leads to a better performance than the RA policy [D. Yueyue 2020], algorithm and learning-agent-based random policy, which reduces the EC. This is because the Deep-RL agent based on the DT algorithm improves the learning accuracy and takes the required action by solving the problem of a huge action-space. Also, it is clear that the reward increases significantly as the assignment time increases.

## 5. Conclusions

In this paper, we proposed the use of sustainable RA- in FL-DT edge networks, which integrates DT into edge networks for real-time data analysis. We also focused on enhancing the security and effectiveness of edge computing in IoT networks, ensuring sustainable computing and analyzing the limitations of the current RA. The B5G networks rely on improvements in the security and effectiveness of edge computing in IoT networks, which require URLLC, real-time data processing, and real-time data analysis to achieve sustainable computing, and mitigate the computation and communication capacity. Moreover, minimizing the cost of system delay and the EC depends on the proposed DT-enabled Deep-RL agent technique for optimal IoT devices to guarantee the efficiency of the learning accuracy and sustainability of the DT-IoT system FL by employing DNN for RA. The simulation results demonstrate that the proposed Deep-RL-agent-based DT can balance system delay and EC to improve reliability by guaranteeing the learning accuracy of IoT devices. This also greatly lowers the system time cost and offers good accuracy, enhancing the efficiency of the integrated DTs with edge networks.

## Figures and Tables

**Figure 1 sensors-23-07262-f001:**
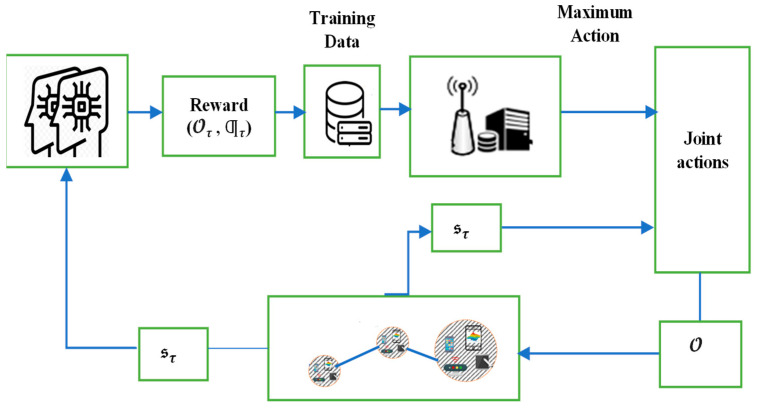
A DT-enabled Deep-RL for edges’ association.

**Figure 2 sensors-23-07262-f002:**
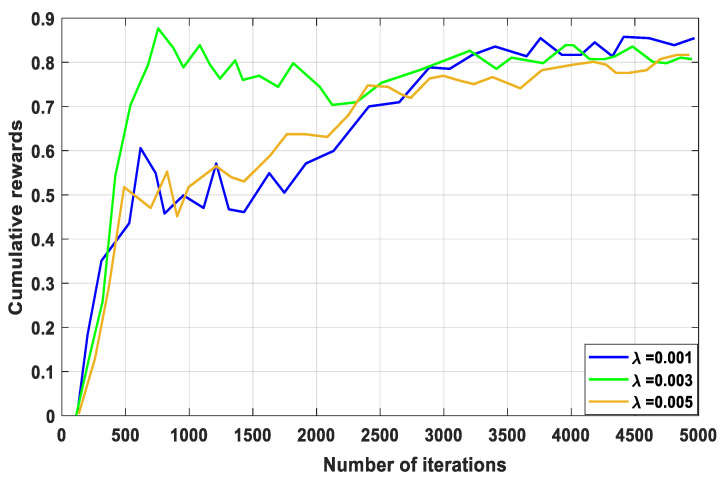
Cumulative reward versus number of iterations.

**Figure 3 sensors-23-07262-f003:**
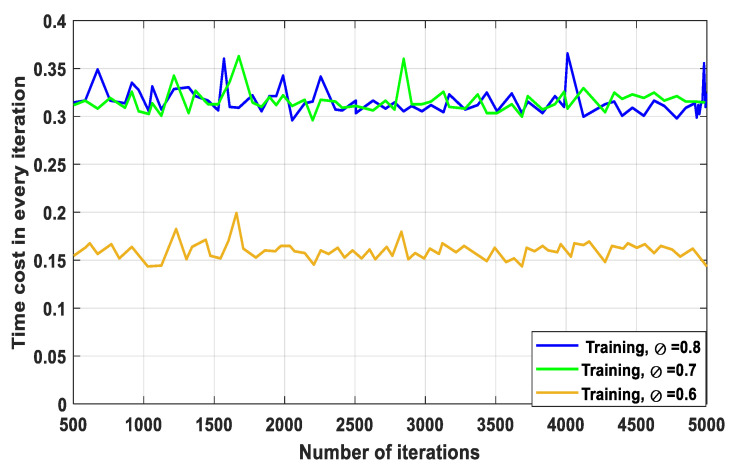
Time cost related to number of iterations.

**Figure 4 sensors-23-07262-f004:**
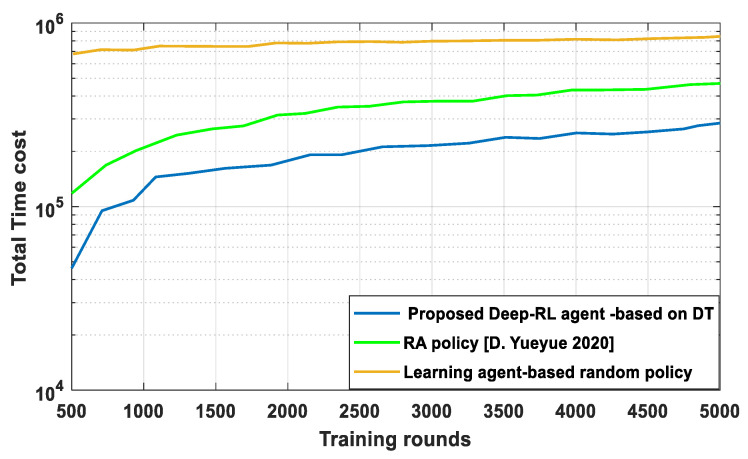
Total time cost versus training round.

**Figure 5 sensors-23-07262-f005:**
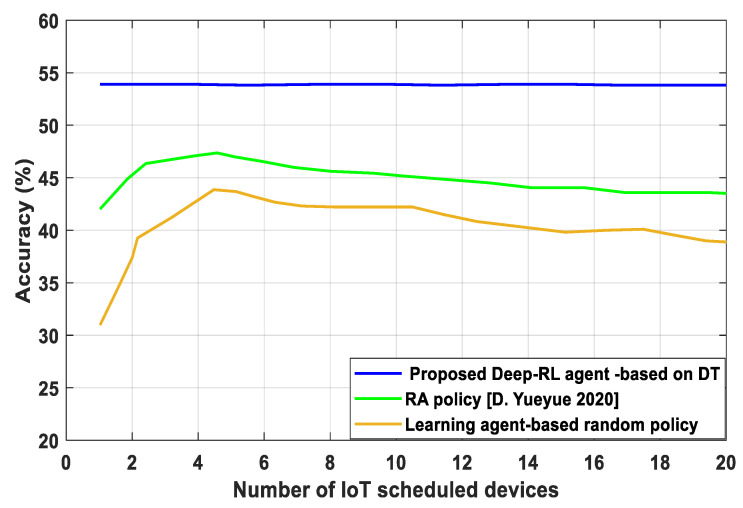
Accuracy versus number of IoT scheduled devices.

**Figure 6 sensors-23-07262-f006:**
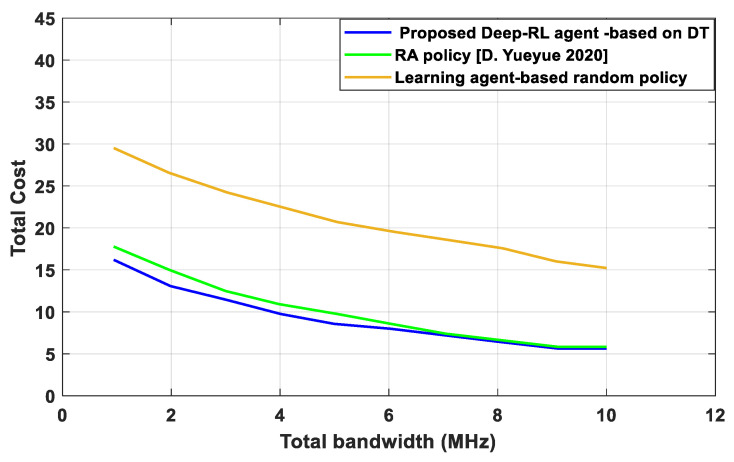
Total cost versus bandwidth.

**Figure 7 sensors-23-07262-f007:**
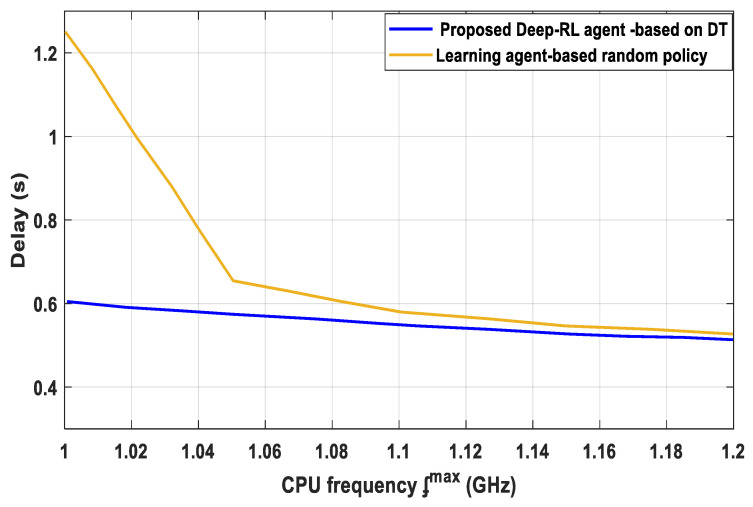
Relay versus CPU frequency.

**Figure 8 sensors-23-07262-f008:**
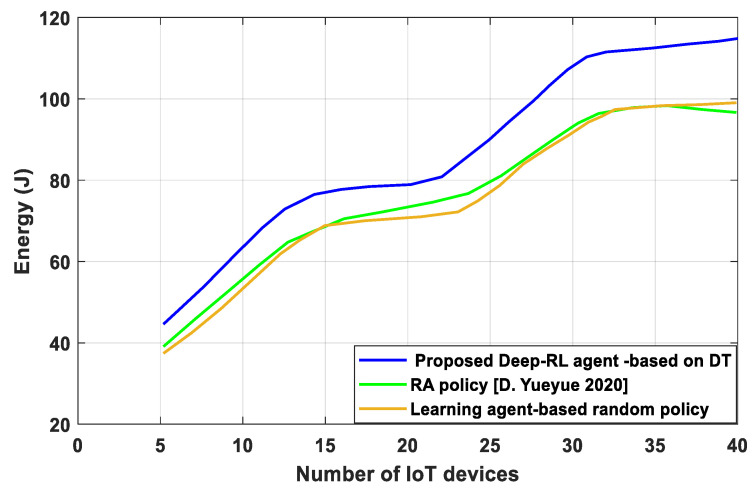
EC versus number of IoT devices.

**Table 1 sensors-23-07262-t001:** Simulation parameters.

Parameter	Value
Communication bandwidth	30 MHz
Transmit power 𝒫	20 dBm
Adjusting the target weight [1,6,12]	0.8
CPU clock speed mobile edge server [17,19]	$1.2\times 10^{12}$ Hz
Discount factor [25,26,27]	0.5
Time slot duration τ [1,3,10]	0.05
Noise figure	5 dB
The data size of each IoT device [25,28]	[0~50] MB

## Data Availability

The original data can be obtained from the open access online dataset from MATLAB Math Work: https://in.mathworks.com/help/deeplearning/gs/create-simple-sequence-classification-network.html.

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
