# Peer review of "Sustainable Resource Allocation and Reduce Latency Based on Federated-Learning-Enabled Digital Twin in IoT Devices"

_sensors, 2023, doi:10.3390/s23167262_

Round 1

Reviewer 1 Report

The work proposed a framework to allocate the bandwidth resource when considering shortening communication and encryption latency and acheving model accuracy.

overall the work is contemporary and potentially has application value.

One aspect that is not expressed clearly is the reward, from the context of description, the formulation is either minimizing the latency or minimizing the training loss/objective. Yet when introducing the Reinforcement learning, the reward is the latency plus or minus a constant C(i.e., the constant cycle time of CPU), for example T+C, . Then, it’s not very clear when first min T and then max T iteratively. Is it essentially become a minimax problem? 

Also it mentioned the reward is Energy in figure 8, but it seems to consider latency in equation (16). Is it changed somewhere in the text? 

Some of the background can be more concise, yet the specific context-related problem settings can be improved. 

Author Response

Reviewer 1

Comment: Comments and Suggestions for Authors

The work proposed a framework to allocate the bandwidth resource when considering shortening communication and encryption latency and achieving model accuracy.

Author action: Thank you so much for your valuable comments to improve our manuscript. We update the manuscript according to your invaluable comments and suggestions. The authors highly appreciated your suggestions and comments for improving the paper.

Comment 1: One aspect that is not expressed clearly is the reward, from the context of description, the formulation is either minimizing the latency or minimizing the training loss/objective. Yet when introducing the Reinforcement learning, the reward is the latency plus or minus a constant C(i.e., the constant cycle time of CPU), for example T+C. Then, it’s not very clear when first min T and then max T iteratively. Is it essentially become a minimax problem?

Author action 1: In page (9); Subsection 3.2. Deep- RL Agent for RA: By using a positive reward  in combination with an active learning technique can greatly reduce the cost of training a model. Based on the current system states obtained from digital twins the performance of the action is quantified by the reward function at the end of the iteration. The constraints in (16) , will also reduce system time costs and every agent attempts to utilize the optimum policy to maximize an accumulative reward based on frequently updated , and  as shown in (19) with minimizing the weighted cost of transmission policy and enhancing learning efficiency at every step.

Comment 2: Also it mentioned the reward is Energy in figure 8, but it seems to consider latency in equation (16). Is it changed somewhere in the text?

Author action 2: In page 15, at Fig. 8. We updated the manuscript by adding explanations.  “To improve learning accuracy, reducing the energy consumption of the DT-IoT system depends on enhancing the target of local computing by decreasing wait times and minimizing the communication load. To achieve the high efficiency of learning accuracy from (13c), (13d), the total energy cost  is also checked as a budget that should not exceed an expected value  the agent continues on to the next state and receives a reward immediately.”

Comment 3: Some of the background can be more concise, yet the specific context-related problem settings can be improved?

Author action 3: We updated the manuscript by adding explanations and improved the problem statements.

Reviewer 2 Report

The authors in this paper work on for dependable real-time data processing, association Digital Twins (DT) with edge networks using blockchain technology. As it also offers a safe, scalable solution to bridge the gap between physical edge networks and digital systems. Authors proposed a blockchain-based Federated Learning (FL) architecture for collaborative computing that utilizes the DT edge network. The proposed architecture improves system security and dependability while increasing data privacy. This depends on a balance between system latency and energy consumption based on the proposed DT-empowered Deep Reinforcement Learning (Deep-RL)-agent in order to provide sustainable Resource Allocation (RA) and ensure real-time data processing interaction between Internet of Things (IoT) devices and edge servers.

The overall comments are mentioned below:

1.       The significant contribution of the paper is not found.

2.       The problem statement and its impact to the various real-life applications are not listed.

3.       Authors written to address the security and privacy challenges using Blockchain and edge computing for IoT devices. The Blockchain is secure, but this security features come with lots of computation power, how authors able to address this computational issue is notwhere mention?

4.       IoT device as well as edge device are resource constraint how Blockchain used in these devices.

5.       Which consensus protocols used to make the secure communication in the proposed system?

6.       For calculation of latency what is the size of data set used?

7.       In proposed algorithm authors refer articles number [16], [18] and [19] , what is authors contribution to this?

8.       Result section need lots of analysis.

9.       The comparative statement is also required to authors proposed method with existing one.

10.   The proposed system security analysis is done by the authors as they claim that using Blockchain it secures the system.

11.   The paper needs to recheck again to avoid typo and grammar error. 

Major revision 

Author Response

Reviewer 2,

Comments and Suggestions for Authors

The authors in this paper work on for dependable real-time data processing, association Digital Twins (DT) with edge networks using blockchain technology. As it also offers a safe, scalable solution to bridge the gap between physical edge networks and digital systems.

Author action: Thank you so much for summarizing our contribution. We have updated the manuscript according to your invaluable comments and suggestions. The authors are highly appreciated your suggestions and comments for improving the paper.

Comment 1:   The significant contribution of the paper is not found.

Author action 1: In Pg. (3), Subsection Related Works. We already update the title according to your comments “The significant contribution of this framework is to establish an enduring RA system, guaranteeing seamless real-time communication between IoT devices and edge servers. This accomplishment hinges on the formulation of an optimized data relay challenge, wherein Deep Neural Networks (DNN) are harnessed as the strategic schedulers within the suggested approach. The objective here is to strike a harmonious equilibrium between system responsiveness and EC, all orchestrated through the innovative utilization of the Deep-RL agent empowered by the proposed DT. The Deep-RL agent evaluates the efficiency of action by taking into account RA actions within the DT. According to iterations and actions taken, this assessment entails allocating bandwidth resources to IoT devices. The main goal is to come up with an ideal policy that increases learning effectiveness at each stage of the procedure.”

Comment 2: The problem statement and its impact to the various real-life applications are not listed.

Author action 2: In Pg. (2), subsection Related Works. We update the manuscript according to your comments. Emerging technologies like mobile edge computing and next-generation connectivity are essential for facilitating the IoT's quick development and adoption. Massive amounts of privacy-sensitive data are produced in sustainable computing networks as a result of the expanding scope of data-driven applications. It's difficult to figure out how to interpret such data on IoT devices with limited resources. The digitalization of the IoT is restricted by limited computing power and communication capabilities, which makes it difficult to adopt blockchain to develop DT models that demand trust and consensus across distributed users. Additionally, blockchain technology enhances distributed learning's effectiveness in solving the problem of data privacy with high-dimensional and time-varying attributes based on the provided Deep-RL.

Comment 3: Authors written to address the security and privacy challenges using Blockchain and edge computing for IoT devices. The Blockchain is secure, but this security features come with lots of computation power, how authors able to address this computational issue is notwhere mention?

Author action 3: We already revised the abstract according to your comments “Concerns about security and privacy could be addressed by integrating blockchain and improving data security, by enabling FL to improve network dependability. This is accomplished by strengthening the resource scheduling algorithm in the DT-IoT framework in order to reduce system delays, improve edge computing capabilities, and guarantee reliable and secure computations in DNNs. A Deep-RL agent that utilizes the DT foundation is introduced by the novel technique. In order to optimize user scheduling and bandwidth allocation among IoT devices, this agent evaluates performance actions using RA. The ultimate goal is to increase system stability as a whole.”

Comment 4: IoT device as well as edge device are resource constraint how Blockchain used in these devices.

Author action 4: In Pg. (7), section 3 Formulation problem. We already update the motivation according to your comments” Motivated by the above issues, the transmission time for uploaded data is determined by a number of  that indicate the behaviour model of IoT devices developed by analyzing their historical operating data throughout time slots provided to IoT devices at iteration by .  The restriction (13c) denotes computational resources assigned for cycle CPU frequency that are fewer than the upper or maximum constraints . In contrast to IoT devices, edge servers from (13d) have a suitable power supply. As seen in (13c) and (13d), when resources are limited, blockchain technology can be included in both edge and IoT devices.”

Comment 5: Which consensus protocols used to make the secure communication in the proposed system?

Author action 5: In Pg. (5), (6), subsection 2.1. Blockchain-enabled FL is capable of increasing reliability and enhancing data security in a network in order to enable secure collaborative learning and foster trust among untrusted users. The proposed method incorporates blockchain and DT based on improving output accuracy and lowering loss to create secure-aware and reliable-aware edge intelligence. Due to the proliferation of user devices, it is necessary to shorten the amount of time needed for model training in the various applications as illustrated in (2) and (3) in order to maintain secure connections.

Comment 6: For calculation of latency what is the size of data set used?

Author action 6: In Pg. (6), subsection 2.1. We update the manuscript according to your comments “Depending on the particular circumstance and the components of latency being assessed, the dataset size utilized for latency calculations in a blockchain setting can change.  Reducing the amount of time needed for model training across a variety of applications due to the growth of user devices and the necessity for ultra-low latency connection in (3). The amount of time it takes to upload data is influenced by the blockchain latency (4) between edge servers and the communication delay for the data size of DT  that reflects the CPU cycle frequency  (5).

Comment 7: In proposed algorithm authors refer articles number [16], [18] and [19], what is authors contribution to this?

Author action 7: Thank you so much for your comments. “Using data and the optimisation aim to reduce system delay and transmission energy cost, [18] and [19] presented a digital twin IoT to achieve reliable real-time and computation efficiency. The authors also incorporate ideas from [16], [18], and [19] to increase the hierarchical DTIoT system's effectiveness, ensure learning accuracy, reliability, and security, and achieve a balance between system latency and consumption of energy.”

  Comment 8: Result section need lots of analysis.

Author action 8: In section Discussion, “The size of the steps taken to update the model parameters during training is determined by the learning step size, which is an important variable shown in Fig. 2. Convergence speed and stability are impacted. A new transaction is added to the blockchain every time a learning step involves changing the learning step size. The iteration number, learning step size, and possibly other pertinent metadata are all recorded in this transaction along with other pertinent information.

- The discount factor in Deep-RL, as seen in Fig. 3, symbolises the preference for future benefits over present ones. Through the blockchain, transactions on a distributed network are verified and agreed upon. Due to the increase in training from  0.6 to  0.8, this process naturally generates time delays. The blockchain has the ability to contribute value in terms of data openness to avoid limiting the effectiveness of Deep-RL training. It is directly able to reduce the time required to train a Deep-RL agent.

Comment 9: The comparative statement is also required to authors proposed method with existing one.

Author action 9: Thank you so much for your comments. In Pg.3, subsection Related Works. “These studies [18-23] did not focus on how to address the RA for the user scheduling, and bandwidth allocation in IoT devices based on Deep-RL for DT to evaluate the performance action. Moreover, the above studies [18-23] have not investigated the sustainable RA, optimized data relay challenge, wherein Deep Neural Networks (DNN) are harnessed as the strategic schedulers within the strategy scheduler in the recommended solution to balance learning accuracy and time expenditure.   This work is different from the previously existing ones. Whereas this work focused on addressing sustainable RA and ensuring real-time data processing interaction between IoT devices and edge servers. The significant contribution of this framework is to establish an enduring RA system, guaranteeing seamless real-time communication between IoT devices and edge servers. This accomplishment hinges on the formulation of an optimized data relay challenge, wherein Deep Neural Networks (DNN) are harnessed as the strategic schedulers within the suggested approach. The objective here is to strike a harmonious equilibrium between system responsiveness and EC, all orchestrated through the innovative utilization of the Deep-RL agent empowered by the proposed DT. The Deep-RL agent evaluates the efficiency of action by taking into account RA actions within the DT. According to iterations and actions taken, this assessment entails allocating bandwidth resources to IoT devices. The main goal is to come up with an ideal policy that increases learning effectiveness at each stage of the procedure.”

Comment 10:  The proposed system security analysis is done by the authors as they claim that using Blockchain it secures the system.

Author action 10: A complete assessment of the system's architecture, components, and integration of blockchain technology to give security benefits is required when performing a system security analysis to determine how blockchain improves security, as demonstrated in [5], [7], [13], [14], [15], [17], [20].

Comment 11: The paper needs to recheck again to avoid typo and grammar error.

Author action 11: Thank you very much for your concern to improve our manuscript. We have check and updated the manuscript. Authors are highly appreciated your suggestions and comments for improving paper.

Round 2

Reviewer 2 Report

Authors are given responses. No more comments from my side.